

**Eurasian autumn snow impact on winter North Atlantic Oscillation**
**depends on cryospheric variability**

Martin Wegmann (1), Marco Rohrer (2,3,*), María Santolaria-Otín (4) and Gerrit Lohmann (1)
(1) Alfred Wegener Institute, Helmholtz Centre for Polar and Marine Research,
Bremerhaven, Germany
(2) Oeschger Centre for Climate Change Research, University of Bern, Bern, Switzerland
(3) Institute of Geography, University of Bern, Bern, Switzerland
(4) Institut des Géosciences de l'Environnement, Université Grenoble-Alpes, France
(*) now at: Axis Capital, Zurich, Switzerland
**Abstract:**
In recent years, many components of the connection between Eurasian autumn snow cover
and wintertime North Atlantic Oscillation (NAO) were investigated, suggesting that
November snow cover distribution has strong prediction power for the upcoming Northern
Hemisphere winter climate. However, non-stationarity of this relationship could impact its
use for prediction routines. Here we use snow products from long-term reanalyses to
investigate interannual and interdecadal links between autumnal snow cover and
atmospheric conditions in winter. We find evidence for a negative NAO tendency after
November with a strong west-to-east snow cover gradient, which is valid throughout the
last 150 years. This correlation is linked with a consistent impact of November snow on a
slowed stratospheric polar vortex. Nevertheless, interdecadal variability for this relationship
shows episodes of decreased correlation power, which co-occur with episodes of low
variability in the November snow index. We find that the same is also true for sea ice as an
NAO predictor. The snow dipole itself is associated with reduced Barents-Kara sea ice
concentration, increased Ural blocking frequency and negative temperature anomalies in
eastern Eurasia. Increased sea ice variability in recent years is linked to increased snow
variability, thus increasing its power in predicting the winter NAO.
**Keywords: SNOW, NAO, SEA ICE, VARIABILITY, PREDICTION**



## 1. Introduction

As the leading climate mode to explain wintertime climate variability over Europe (**Thompson and Wallace 1998**), the North Atlantic Oscillation (NAO) has been extensively studied over the last decades (**Wanner et al. 2001, Hurrell and Deser 2010, Moore and Renfrew 2012, Pedersen et al. 2016, Deser et al. 2017**). The NAO has been defined as the strength of the pressure gradient between Iceland (representing the edge of the polar front) and the Azores (representing the subtropical high ridge). The sign of the NAO is related to weather and climate patterns stretching from local to continental scales. Since its configuration has severe socioeconomic, ecological and hydrological impacts for adjacent continents, seasonal to decadal predictions of the state of the winter NAO are high priority research for many climate science centers (**Jung et al. 2011, Kang et al. 2014, Scaife et al. 2014, Scaife et al. 2016, Smith et al. 2016, Dunstone et al. 2016, Wang et al. 2017, Athanasiadis et al. 2017**).

Together with the rapid warming of the Arctic and the increased frequency of severe winters over Eurasia and North America (**Yao et al. 2017, Cohen et al. 2018, Kretschmer et al. 2018, Overland and Wang 2018**), recent studies highlighted the state of the Northern Hemispheric cryosphere as a useful predictor for the boreal wintertime (DJF) NAO (**Cohen et al. 2007, Cohen et al. 2014, Vihma 2014, Garcia-Serrano et al. 2015, Cohen 2016, Orsolini et al. 2016, Crasemann et al. 2017, Warner 2018**). Although both systems seem to be connected (**Cohen et al. 2014, Furtado et al. 2016, Gastineau et al. 2017**), the emerging main hypothesis connects reduced autumn Barents-Kara sea ice concentration and increased Siberian snow cover with a negative NAO state in the following winter months (**Cohen et al. 2014**).

The proposed mechanism behind this hypothesis is a multi-step process, starting with autumn sea ice loss for the European Arctic, followed by altered tropospheric circulation due to elevated Rossby wave numbers, vertical propagation of said Rossby waves upward into the stratosphere and consequently a weakening of the polar vortex (see **Cohen et al. 2014** for an in depth discussion). With the weakening (or the reversal) of the polar vortex, a stratospheric warming signal manifests. This signal propagates slowly back into the troposphere, where it is expressed as a negative NAO, connected to the concurrent cold winters for Eurasia (**Kretschmer et al. 2018**).

In recent years, many components of this pathway were investigated, especially concerning the increased frequency of cold winters over Europe and the emergence of the counter-intuitive "Warm Arctic – cold continent" (WACC) pattern over Eurasia (**Petoukhov and Semenov**



**2010, Vihma 2014**). However, there remains substantial uncertainty about the impact of Arctic
sea ice in terms of location (**Zhang et al 2016, Luo et al 2017, Screen 2017, Kelleher and**
**Screen 2018**), timing (**Honda et al 2009, Overland et al 2011, Inoue et al 2012, Suo et al**
**2016, , Sorokina et al 2016, King et al 2016, Screen 2017, Wegmann et al 2018a, Blackport**
**and Screen 2019**) or if sea ice can be used as a predictor/forcing at all based on the contrasting
result of model studies (**McCusker et al. 2016, Collow et al. 2016, Pedersen et al. 2016,**
**Boland et al. 2017, Crasemann et al. 2017, Ruggieri et al. 2017, Garcia-Serrano et al. 2017,**
**Francis 2017, Screen et al. 2018, Mori et al. 2019, Hoshi et al. 2019, Blackport et al. 2019,**
**Romanowksy et al. 2019**).
The interplay between Arctic sea ice and Siberian snow is much less explored. **Ghatak et al.**
**(2010)** showed that reduced autumn polar sea ice leads to the emergence of increased Siberian
winter snow cover, especially so in the eastern part of Eurasia. This dipole signal was amplified
in coupled climate model runs for the 21$^{st}$ century, where sea ice is substantially diminished. In
an observational study, **Yeo et al. (2016)** point out that the moisture influx from the open Arctic
ocean into the Eurasian continent contributes to the increase of snow cover, a mechanism that
**Wegmann et al. (2015)** describe. **Gastineau et al. (2017)** found that reduced sea ice is
connected to a distinct November snow dipole over Eurasia, both in reanalysis and model data.
They further state, that the snow component is a statistically more powerful predictor for the
atmosphere in the following winter. This relationship was also found in a range of climate
models, albeit with weaker links. **Xu et al. (2019)** found the same correlation in observational
and model data, however looking at winter climate only. Based on their analysis, the authors
state that the enhanced snow cover in winter is a product of the negative NAO rather than a
precursor. **Sun et al. (2019)** highlight the importance of elevated North Atlantic sea surface
temperatures for the development of a Eurasian snow dipole in autumn. This warming of the
North Atlantic favors increased Rossby wave numbers, eventually forming a high pressure
anomaly over the Ural Mountains, transporting cold air masses towards the south of its eastern
flank.
The possible impact of the Siberian snow on the stratosphere and eventually on the NAO is
well summarized in **Henderson et al. (2018)**. Although observational NAO prediction studies
with Siberian snow showed great success in the past (**Cohen and Entekhabi 1999, Saito et al.**
**2001, Cohen et al. 2007, Cohen et al. 2014, Han and Sun 2018**), links between snow and the
stratosphere still seems to be missing or too weak in model studies (**Furtado et al. 2015,**
**Handorf et al. 2015, Tyrrell et al. 2018, Gastineau et al. 2017, Peings et al. 2017**), whereas





nudging realistic snow changes to high resolution models seems to improve the prediction
power (**Orsolini and Kvamsto 2009, Orsolini et al. 2016, Tyrrell et al. 2019**). Moreover,
even though the stratosphere–surface connection is now reasonably well established
(**Kretschmer et al. 2018**), the timing and location of the snow cover used for the prediction is,
as with sea ice, still debated (**Yeo et al. 2016, Gastineau et al. 2017**). As an additional caveat,
**Peings et al. (2013)** and more recently **Douville et al. (2017)**, showed that the proposed autumn
snow-to-winter NAO relationship is non-stationary for the 20$^{th}$ century. A possible modulator
for that relationship might be the phase of the Quasi Biennial Oscillation (QBO) (**Tyrrell et al.**
**2018, Peings et al. 2017, Douville et al. 2017**). **Peings (2019)** argues that neither snow nor sea
ice anomalies trigger the stratospheric conditions needed to produce winter extremes and that
instead high tropospheric blocking frequency over Northern Europe leads to the cryosphere
anomalies.
Here, we follow up on the consequences of recent studies (**Han and Sun 2018**, **Gastineau et**
**al. 2017)** who point out the predictor strength of the November snow cover dipole for the
following winter month, to revisit the question of a) non-stationarity in the 20$^{th}$ century, b)
importance of snow versus sea ice as predictor and c) possible precursors/modulators of the sea
ice–snow–stratosphere link. With this we aim to contribute to the understanding of impacts of
cryosphere variability on midlatitude circulation (**Francis 2017, Henderson et al. 2018,**
**Blackport et al. 2019**). To this end, we utilize centennial reanalyses and reconstruction data,
where we focus on the transition from October to November to DJF to facilitate the idea of
seasonal prediction.
This paper is organized as follows: Section 2 describes the data and methods used. In section
3, we introduce the snow cover indices and their interannual prediction value. Section 4
investigates interdecadal shifts in the correlation between snow cover and NAO as well as
possible determining factors. The results are discussed in section 5 and finally summarized in
section 6.
2. **Data and Methods**

a.   Atmospheric reanalyses

To evaluate long-term reanalyses, we use snow cover, snow depth and atmospheric properties
from the MERRA2 reanalysis (**Gelaro et al. 2017**). MERRA2 has a dedicated land surface
module and was found to reproduce local in-situ snow conditions over Russia very well



(**Wegmann et al. 2018b**). For an in-detail description of how MERRA2 computes snow
properties see e.g. **Orsolini et al. (2019)**.
To cover the 20th century and beyond, we include two long-term reanalyses in this study,
namely the NOAA-CIRES 20th century reanalysis Version 2c (20CRv2c) (**Cram et al. 2015**)
as well as the Centre for Medium-Range Weather Forecasts (ECMWF) product ERA-20C (**Poli**
**et al. 2016).** From the ERA-20C product we use snow depth, whereas from 20CRv2c we
investigate snow depth and snow cover. Both reanalyses were found to represent interannual
snow variations over Eurasia remarkably well. For an in-depth discussion of their performance
and their technical details concerning snow computation see **Wegmann et al. (2017)**. We also
performed the same analysis using the coupled ECMWF reanalysis CERA-20C (**Laloyaux et**
**al., 2018**), but found no added knowledge gain over ERA-20C. Thus, we do not include CERA-
20C in any further analysis.
We use these three reanalysis products to extend the October and November index proposed by
**Han and Sun (2018)** into the past, where the November index is in essence the snow dipole
found by **Gastineau et al. (2017)** using maximum covariance analysis (Figure 1). Where the
October index is just calculated as field average snow cover, the November index is computed
as difference between the eastern and the western field average. It should be noted, that **Han**
**and Sun (2018)** found the November index to be linked to a negative NAO and colder Eurasian
near-surface temperatures, whereas the October index was correlated with warmer-than-usual
temperatures over Eurasia and a southward-shifted positive DJF NAO. However, since many
studies focus on Northern Eurasian October snow cover as the predictor for winter climate, we
will include it nonetheless. MERRA2 and 20CRv2c offer snow cover as well as snow depth as
a post-process output, however ERA-20C only offers snow depth. We refrain from converting
it to snow cover ourselves, but found the index based on snow depth to be extremely similar
(also see Supplementary Figure 1) to the same index using snow cover. All snow indices are
normalized and linearly detrended with respect to their overall time period. Generally, we found
the long term reanalyses to be of comparable quality of MERRA2 during the overlapping
periods.





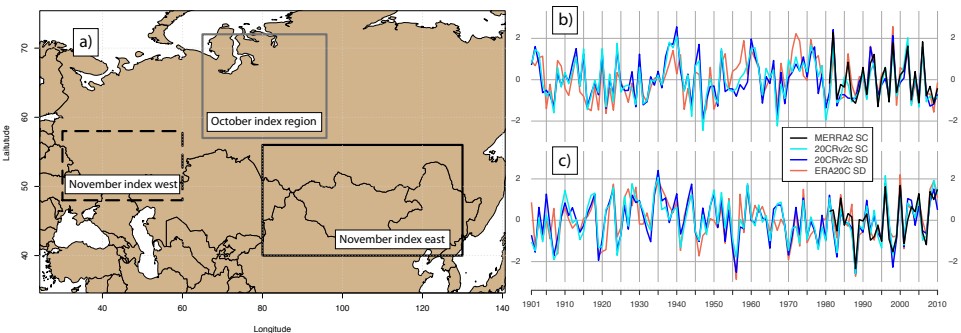


Figure 1: a) Regions for October and November snow index used in this study. b) Linearly detrended and normalized October snow index comparison for the 20th century for snow cover (SC) and snow depth (SD) variables. c) same as b) but for the November snow dipole.

Besides snow properties we use atmospheric and near-surface fields from all three reanalyses. Moreover, as **Douville et al. (2017)**, we use the field averaged (60°–90° N) 10 hectopascal (hPa) geopotential height (GPH) anomalies in ERA-20C as a surrogate for polar vortex (PV) strength. Although ERA-20C only assimilates surface pressure, correlation between this stratospheric index in ERA-20C and MERRA2 during the overlapping time periods is above 0.9.

The ERA20C 10 hPa November–December mean GPH shows remarkable interannual agreement with state-of-the-art reanalyses that assimilate upper air data for the period 1958–2010 (see Supplementary Figure 2). Moreover, MERRA2 and ERA20C 10 hPa GPH anomalies agree best over the northern polar regions with correlation coefficients of >0.9 for the period 1981–2010 (see Supplementary Figure 2). This fact supports the extended value of the ERA20C polar stratosphere. Before 1958, the quality of the ERA20C stratosphere is difficult to assess, but the comparison with reconstructions of 100 hPa GPH zonal means shows very good agreement for late autumn and winter months (see Supplementary Figure 3). As the 20CRv2c ensemble mean dilutes the interannual variability signal back in time with increased variability within the ensemble members, we use the deterministic run of ERA20C for the following stratosphere analyses.

We use 6-hourly 500 hPa GPH fields (GPH500) to calculate monthly blocking frequencies according to **Rohrer et al. (2018)**. Blockings are computed according to the approach introduced by **Tibaldi and Molteni (1990)** and are defined as reversals of the meridional GPH500 gradient. In accordance to **Scherrer et al. (2006)** the one-dimensional **Tibaldi and**



**Molteni (1990**) algorithm is extended to the second dimension by varying the latitude between
35° and 75° instead of a fixed latitude:
i) GPH500 gradient towards pole: $GPH500 G_P = \frac{GPH500_{\varphi+d\varphi} - GPH500_\varphi}{d\varphi} < -10 \frac{m}{°lat}$ (1)

ii) GPH500 gradient towards equator: $GPH500 G_E = \frac{GPH500_\varphi - GPH500_{\varphi-d\varphi}}{d\varphi} > 0 \frac{m}{°lat}$ (2)

Blocks by definition are persistent and quasi-stationary high-pressure systems that divert the
usually prevailing westerly winds in the mid-latitudes. They influence regional temperature and
precipitation patterns for an extended period. Therefore, not all blocks that fulfill above
mentioned two conditions are retained. We only include blocks that have a minimum required
lifetime of 5 days and a minimum overlap of the blocked area of 70% ($A_{t+1} \cap A_t > 0.7 * A_t$)
in our blocking catalog. This largely follows the criteria defined by **Schwierz et al. (2004)**.
b) Climate reconstructions
To be as independent as possible with regards to the reanalyses we use a wide array of climate
index reconstructions for the 20[th] century:
• Atlantic Multidecadal Oscillation (AMO): For the AMO index we take October values
based on the **Enfield et al. (2003)** study. We choose October to allow for a certain
feedback lag with the atmosphere and to have decent prediction value for the upcoming
snow and NAO indices.
• El Niño – Southern Oscillation (ENSO): We chose the ENSO3.4 reconstruction based
on the HadISSTv1 **Rayner et al. (2003)** SSTs. As with the AMO, we select October
values to allow for a reaction time in the teleconnections.
• North Atlantic Oscillation (NAO): We use the extended **Jones et al. (1997)** NAO index
for DJF from the Climate Research Unit (CRU).
• Sea Ice: We use the monthly sea ice reconstruction by **Walsh et al. (2017)** which covers
the period 1850–2013 to create a Barents-Kara (65–85°N, 30–90°E) sea ice index for
November.
3. **Results**

a. Interannual links





In the following paragraphs we investigate the year-to-year relationship between the snow
indices and the following winter SLP fields. For this we use MERRA2 for a 35-year-long
window ranging from 1981–2015, ERA20C for a 110-year-long window ranging from 1901–
2010 and 20CRv2c for a 160-year-long window ranging from 1851–2010.
Figure 2 shows the linear regression fields of DJF SLP anomalies projected onto the respective
snow indices in October and November. For October, we find no NAO-like pressure anomaly
appears to be significantly correlated with the snow index in any of the three reanalysis products
and respective time windows (Figure 2a,b,c). Instead, negative SLP anomalies dominate
Northern Eurasia in MERRA2, with high pressure anomalies towards the Himalayan Plateau.
The 110-year-long regression in ERA20C shows significant negative anomalies over the Asian
part of Russia, reaching as far south as Beijing. A second significant negative SLP pattern
appears along the Pacific coast of Canada. Finally, SLP anomalies in 20CRv2c support the main
SLP patterns shown by ERA20C, but reduce the extent of negative anomalies over Eurasia and
increase the extent of the negative anomalies over the North Pacific.
The DJF SLP anomaly patterns change substantially when projected onto the November snow
index (Figure 2d,e,f). All three reanalysis products show negative NAO-like pressure anomalies
with significantly positive anomalies over Iceland and the northern North Atlantic and
significantly negative anomalies south of ca. 45° N, including Portugal and the Azores. As
expected, MERRA2 shows the strongest anomalies due to the shorter regression period,
however interestingly ERA20C, with the 110-year long analysis period, shows less large-scale
significance for positive anomalies in high latitudes compared to the 150-year-long
investigation period in 20CRv2c (even though non-significant anomalies cover roughly the
same area as in 20CRv2c (not shown)). This can hint towards decadal variations in the strength
of the regression, but could also be due to biases in the reanalyses.
To check for such biases we compared all reanalyses with the SLP reconstruction dataset
HadSLP2r (**Allen and Ansell 2006**), and found that for the regression analysis using the time
period 1901–2010, 20CRv2c overestimates the polar sea level pressure response, whereas
ERA20C is much closer to HadSLP2r (See Supplement Figure 4). This would indeed support
the notion of decadal variations in the strength of the relationship between predictor and
predictand. However, it is worth highlighting that this overestimation for 20CRv2c is not visible
for the 1851–2010 period, where the regression anomalies resemble HadSLP2r much closer.



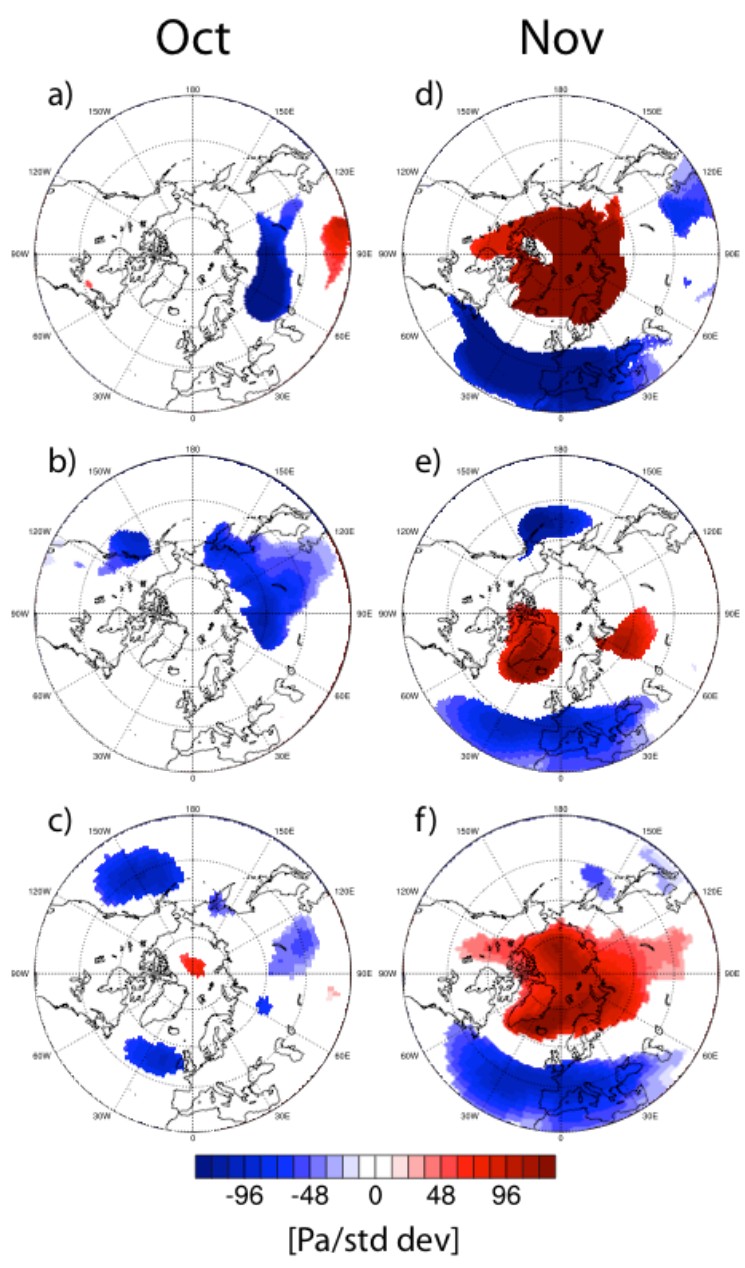

Figure 2: DJF sea level pressure [Pa/std dev] anomalies projected onto snow indices in October (left) and November (right)
for aandd) MERRA2 covering 1981–2015, bande) ERA20C covering 1901–2010 and candf) 20CRv2c covering 1851–2010.
Only anomalies >95% significance level are shown.

We investigate other possible predictors for wintertime NAO via regressed anomalies onto the

November Barents-Kara-Sea (BKS) ice concentration, November–December mean polar GPH



at 10 hPa, October AMO and October ENSO indices (Figure 3). The periods for MERRA2 and
ERA20C are identical as for Figure 2, whereas the anomaly plots for 20CRv2c are using the
maximum period covered in the reconstructions, namely 1851–2010 in the sea ice
reconstruction, 1856–2010 in the AMO reconstruction, 1901–2010 for the polar 10 hPa GPH
index taken from ERA20C, and 1870–2010 for the ENSO reconstruction.
As can be seen from Figure 3, the 35-year-long analysis in MERRA2 shows November sea ice
concentration and early winter stratospheric heights to regress a similar SLP pattern than the
November snow index. Positive SLP anomalies over Iceland and Greenland combined with
negative anomalies over Southern Europe and the adjacent North Atlantic shape a negative
NAO-like pattern in DJF (Figure 3a). On the other hand, the interannual signals in the October
AMO and ENSO indices do not point towards such a pressure distribution. The small
interannual changes and low frequency of the AMO combined with the short sample period
prohibit most of the significance, only Southern Eurasia shows regions with elevated SLP.
Anomalies regressed on the ENSO index show, as expected, significance mostly for the North
Pacific and North American region.
Looking at the regression patterns in the centennial reanalyses, the NAO-like pattern in the SLP
anomalies regressed onto sea ice and stratospheric GPH can still be seen, however the extent
and strength is substantially reduced compared to MERRA2 as well as compared to the
regression using November snow as predictor. Again, ERA20C shows a decrease in the
significant anomalies regressed onto sea ice compared to 20CRv2c, with possible reasons
already discussed above. Elevated geopotential heights at 10 hPa consistently increases polar
sea level pressure in the following winter months, however the impact over the European and
North Atlantic domain is weak.
SLP anomalies regressed onto the AMO index show significant positive SLP regions for large
parts of Eurasia as well as positive anomalies over the North Atlantic west of Great Britain.
Interesting to note in 20CRv2c is the very strong high-pressure anomaly reaching from the BKS
to the southern part of the Ural mountains, a prominent feature often found for years with
positive AMO and negative sea ice concentration, frequently linked to a high frequency of Ural
blockings (UBs). SLP distribution after El Niño events does not change considerably
irrespective of the dataset and time period used. A strong Pacific signal shows the northern part
of the Pacific-North American pattern (PNA) with negative SLP anomalies over the eastern
North Pacific.




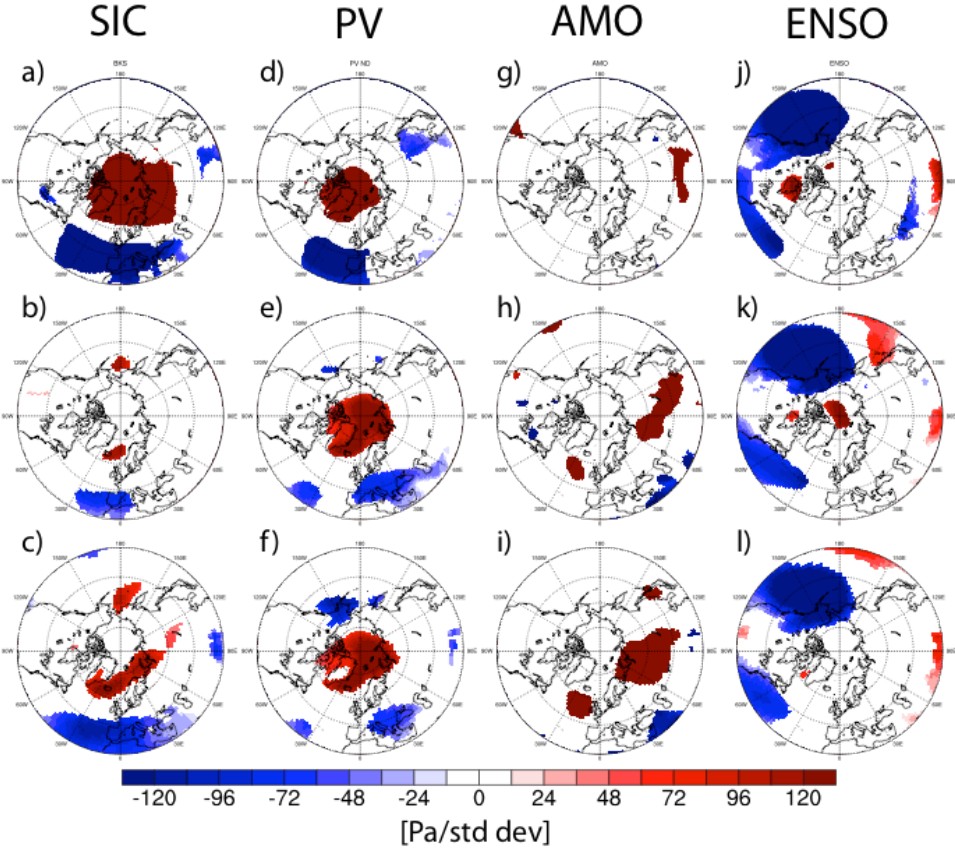


*Figure 3: DJF sea level pressure [Pa/std dev] anomalies projected onto BKS ice concentration in November (far left), polar 10 hPa GPH November December mean (left), October AMO (right) and October ENSO indices (far right) for adgj) MERRA2 covering 1981–2015, behk) ERA20C covering 1901–2010 and cfil) 20CRv2c covering 1851–2010. Regression values for BKS ice concentrations were multiplied by minus one to increase comparability. Only anomalies >95% significance level are shown.*

To investigate the vertical development of climate anomalies connected with the November snow dipole, Figure 4 shows the zonal mean anomalies of zonal wind and temperature in ERA20C projected onto the ERA20C November snow index (for an evaluation with an upper-air reconstruction see supplementary Figure 5). The temporal evolution of the anomalies ranging from October to February shows that stratospheric warming occurs simultaneously within the same month as a positive snow cover dipole, with no stratospheric warming leading that development. Instead, significant surface warming is shown between 60°–90°N for October. The warming signal then dominates the stratosphere and upper troposphere in December, after which the strongest anomalies subside into the lower stratosphere and tropopause in January and February. This development of atmospheric temperatures is mirrored





in the evolution of the polar vortex, where a reduction of the polar vortex and strengthening of
the subtropical jet is seen together with the emergence of the November snow dipole, after
which the region of strongest anomalies migrates from the upper stratosphere to the upper
troposphere.





*Figure 4: Zonal mean (180°E–180°W, 15°N–90°N) left) temperature anomalies and right) zonal mean zonal wind anomalies*
*projected onto snow indices in November for ERA20C covering 1901–2010. Shading indicates 95% significance level.*
To address the physical reasons as to how the low sea ice and high snow indices are connected,
climate anomalies regressed onto BKS ice concentrations for November (Figure 5). Compared
to factors such as AMO and ENSO, BKS sea ice shows a distinct snow cover dipole coinciding
with a high-pressure anomaly over the BKS and the northern Ural mountains, which supports
a regional atmospheric blocking and a cold air advection on its eastern flank. This cold air
anomaly is able to support the snow cover over eastern Eurasia, while relatively warm
temperatures reduce the snow cover over eastern Europe. It should be noted that October BKS
ice concentration shows qualitatively the same pattern for November snow cover anomalies
(not shown), however not statistically significant.

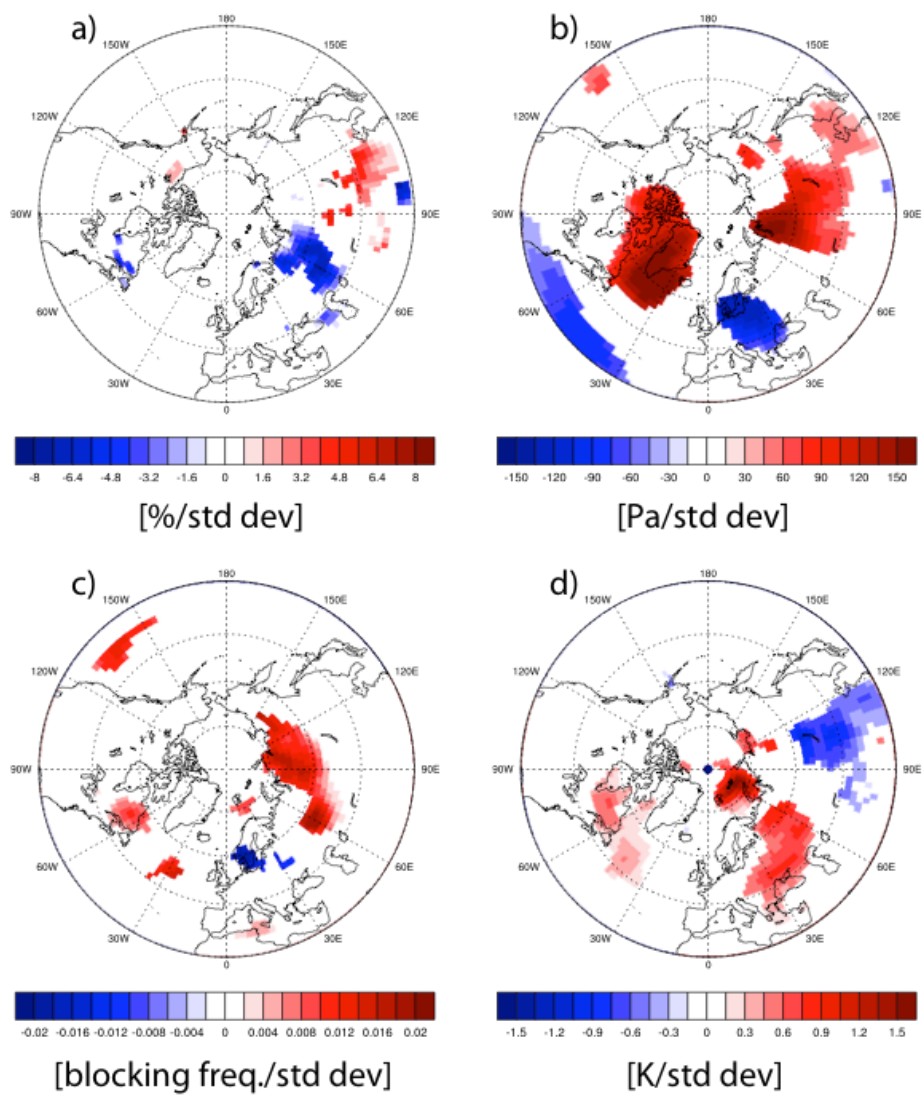

*Figure 5: 20CRv2c November anomalies projected onto BKS ice concentration in November covering 1851–2010. a) November snow cover [%/std dev] anomalies projected onto BKS ice concentration in November, b) November SLP [PA/std dev] anomalies projected onto BKS ice concentration in November, c) November atmospheric blocking [blocking per time unit/std dev] anomalies projected onto BKS ice concentration in November and d) November 2m temperature [K/std dev] anomalies projected onto BKS ice concentration in November. Only anomalies >95% significance level are shown.*

    b.   Interdecadal links

The interdecadal evolution of the November snow index is shown in Figure 6. 21-year running means of the normalized time series of ENSO, AMO, BKS ice and snow hint towards a multidecadal frequency, similar in wave length to the AMO and BKS ice anomalies. Even though we refrain from correlating these time series due to the the 21-year filter (**Trenary and**



**DelSole, 2016**), we find the possible mechanism behind it, which was outlined in the previous
section, to be physically plausible. As **Luo et al. (2016)** point out, warm North Atlantic water
reduces the BKS ice concentration which in turn favors the formation of high pressure over the
Ural mountains and with that, cold air advection towards eastern Eurasia. It should be noted
however, that the AMO and the November snow index are slightly out-of-phase between 1880
and 1920. Possible reasons for that will be examined in the discussion section, but we want to
highlight the strong La Niña events at the end of the 19th century as well as the strong El Niño
events between 1920–1940 that seem to have enough power to influence European climate
(**Brönnimann et al. 2004, Brönnimann et al. 2007, Domeisen et al. 2018**).

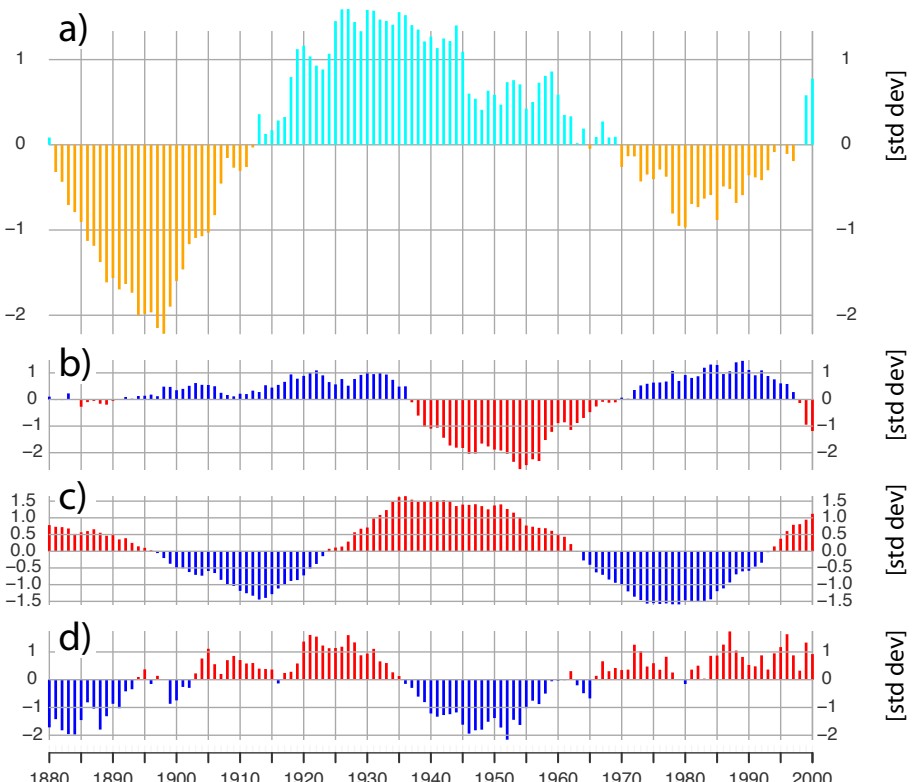


*Figure 6: 21-year running means of a) November snow index from 20CRv2c, b) November BKS ice concentration, c) October*
*AMO and d) October ENSO reconstruction.*

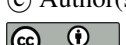

The more critical question is the interdecadal evolution of the relationship between the predictor
and the predictand. Similar to **Peings et al. (2013)** and **Douville et al. (2017)**, we apply a 21-
year running correlation covering the period 1901–2010 to examine the stationarity of the
relationship and differences between 20CRv2c and ERA20C.
Figure 7 summarizes the correlation over time for multiple pairs of climate variables. As Figure
7b points out, the sign of the November snow to winter NAO relationship in 20CRv2c is
negative throughout the whole 20th century. Periods with negative correlations can be found at
the beginning and the end of the century, with relatively weak correlation during the 1930s and
1970s. In ERA20C, these periods are actually marked by positive correlations, indicating a non-
stationary relationship between these two variables. Even stronger decadal variability can be
seen for the running correlations between the October snow index and winter NAO tendency
(Figure 7a), with periods of pronounced negative correlations during the early 20th century
Arctic warming and the 1980s. A recently emerging relationship can be seen in Figure 7c
between BKS ice reduction and the formation of a negative NAO signal in the following winter.
Even though the beginning of the 20th century showed the same sign for this correlation,
decades of positive correlation follow up and last until the late 1970s.
Together with the emergence of the sea ice to NAO relationship, the negative correlations
between BKS sea ice and November snow index as well as between stratospheric warming and
winter NAO strengthen towards the end of the 20th century. This strengthening is also found in
ERA20C for the correlation between November snow and a following stratospheric warming,
where 20CRv2c shows consistently positive correlation values throughout the 20th century.
Overall, the 20CRv2c November snow index shows a more stationary relationship with
tropospheric and stratospheric winter circulation than ERA20C. Possible explanations for this
behavior will be discussed in the following section.




*Figure 7: 21-year centered running correlation time series between a) October snow index and DJF NAO, b) November snow index and DJF NAO, c) November snow index and mean November December polar 10 hPa GPH index, d) November snow index and November BKS ice concentration, e) November BKS ice concentration\*-1 and DJF NAO and f) mean November December polar 10 hPa GPH and DJF NAO index. Black dashed line indicating the 95% confidence level for a two-sided students T-test assuming independence and normal distribution.*

4. **Discussion**



We used a variety of reanalyses and reconstructions to address some of the open questions
regarding the relationship between Eurasian snow cover and the state of the NAO in the
following winter. We followed up on the findings of **Gastineau et al. (2017)** as well as **Han**
**and Sun (2018)** who pointed out the distinct November Eurasian snow cover dipole pattern
and its prediction skill for the winter NAO during recent decades in reanalyses. Given the
importance for seasonal prediction, we address the question of stationarity of said relationship
as well as its context within other popular Northern Hemispheric predictors by utilizing
centennial snow cover, SST and sea ice concentration indices.
Investigating interannual relationships, we could show that the October snow index shows no
skill in predicting the NAO state whereas the November snow index, representing the west-to-
east gradient of snow cover, shows a strong negative statistical relationship for the following
winter NAO. Our findings also support results shown by **Gastineau et al. (2017)**, indicating
that reduced BKS sea ice shows a similar response in DJF SLP anomalies, however its statistical
importance, and therefore quality as being the prime predictor, is less than the November snow
index (see Supplementary Table 1 for partial correlations). SST indices with low interannual
variability, such as AMO and ENSO indices, did not correlate with the DJF NAO state.
The question remained, how stationary this relationship between Eurasian snow and the
wintertime NAO is over time. **Peings et al. (2013)** and the follow up study by **Douville et al.**
**(2017)** found that the October and October–November mean snow cover over a broader region
of Northern Eurasia, and its relationship to the wintertime NAO is indeed not stationary over
time. We could show that by using the November snow dipole, we could extend the results of
**Han and Sun (2018)** and display a very significant (95% significance and higher) negative
interannual correlation with wintertime NAO, going beyond the satellite era up until the mid-
19[th] century. Moreover, we highlight the strong correlation between November snow and
stratospheric warming, supporting the general idea of the physical mechanism proposed by
**Cohen et al. (2014)** and supporting the recent findings of **Gastineau et al. (2017)** and **Douville**
**et al. (2017)**.
In accordance to **Sun et al. (2019)**, decadal variability of the November snow cover index seems
mostly dominated by low frequency variability in the AMO and subsequently reduced or
increased polar sea ice concentration. This mechanism is also supported by the results of **Luo**
**et al. (2016)**, who highlighted the decadal relationship between a positive AMO, reduced sea
ice and increased Ural blocking. Looking at this mechanism on an interannual basis, we show



a robust strengthening of the November snow dipole with decreasing BKS ice concentration, circulation changes over the BKS region and consequently cold air advection towards the eastern part of the snow dipole region. With this, our results support recent studies, which point out the counterintuitive mechanism of Arctic warming and increased continental snow cover via sea ice reduction and circulation changes (**Cohen et al. 2014, Wegmann et al. 2015, Yeo et al. 2016, Gastineau et al. 2017**).

**Peings (2019)** performed model experiments with nudged November Ural blocking fields, BKS ice and snow anomalies. The author found that UB events are not triggered by reduced sea ice, but in fact lead sea ice decrease. Moreover, more November snow alone did not lead to an increase in blocking frequency, nor to a stratospheric warming. The study highlights the UB events as primary predictor for a negative NAO and the Warm Arctic-cold Continents (WACC) pattern. On the other hand, **Luo et al. (2019)** established a causal chain from reduced sea ice to reduced potential vorticity gradient and increased blocking events leading to cold extremes over Eurasia. We computed the field average of blocking frequency within the domain of **Peings (2019)** (10°W-80°E, 45-80°N) and could find a strong correlation with the WACC pattern over time, however only for DJF blocking events (not shown). We find a correlation of November UB events with wintertime NAO, which is however still weaker than the relationship with the November snow dipole, as well as our BKS ice index (see Supplementary Figure 6). Moreover, blockings within the domain of **Peings (2019)** (10°W-80°E, 45-80°N) are not related to a snow dipole whatsoever, neither in October nor in November (see Supplementary Figure 6). That said, we want to highlight the fact that the blocking pattern emerging in Figure 6 is mostly outside of the boundaries of this UB index (10°W-80°E, 45-80°N), and thus might not be caught by this recent study. Furthermore, **Peings (2019)** applies a very general snow cover increase in his nudging experiment, rather than a snow dipole with a west to east gradient.

We also want to point out the possibility of ENSO contributing to a decadal November snow signal. Strong La Niña events at the end of the 19[th] century as well as the strong El Niño events between 1920–1940 seem to extend the periods of low and high snow beyond the frequency of AMO. It is known that strong ENSO events that seem to have enough power to significantly influence European climate (**Domeisen et al. 2018**) on an interannual basis. Moreover, recent studies point towards possible links between the ENSO region, the Madden-Julian Oscillation (MJO) and European climate (**Kang and Tziperman 2017, Garfinkel et al. 2018**). However, more research is needed to make more confident statements about such teleconnections.



**Kolstad and Screen 2019** highlighted the importance of non-stationarity regarding Arctic sea ice and mid-latitude climate variability. In our analysis, running correlations show interdecadal variations concerning the strengths of November snow as the predictor for wintertime NAO. Compared to the analysis of **Douville et al. (2017)**, we could strengthen the stationarity by facilitating the November dipole snow, especially using 20CRv2c snow cover data, but the question behind decadal peaks and valleys in the running correlation persist. To begin with, the October and November snow indices show very different, nearly anticorrelated, running correlation patterns. A high November snow index seems to be a strong predictor for a negative NAO state at the beginning and end of the 20th century, as well as around the 1950s and 60s. On the other hand, the October index, and related to that the results by **Douville et al. (2017)**, shows negative correlations during the 1940s and 1980s.

We found the main reason for the reduction in correlation strength to be reduced variance of the snow index time series during said time period (Figure 8). The reduction of variance is even stronger in ERA20C than in 20CRv2c, which would explain the less stationary correlations in ERA20C. Furthermore, such periods of low snow variability coincide with a reduction of polar vortex variability, hinting even more so towards possible links between November snow and stratospheric temperatures in the following month.

Overall, we advocate the importance of the signal-to-noise ratio rather than mean states for the evolution of the November snow to winter NAO relationship. The general physical link seems rather stable through time, but can be amplified (and dampened) by strong (weak) interannual variability. What exactly causes the snow variability to drop is difficult to assess, but preliminary results support the notion of low variability in AMO and sea ice concentration to have an influence, which supports the above outlined mechanisms (not shown).

Another source of uncertainty is the disagreement between ERA20C and 20CRv2c when it comes to the stationarity of the relationship. 20CRv2c shows negative correlation throughout the whole 20th century, whereas ERA20C flips the sign of the correlation in the late 1930s and late 1970s. The same relationship but using October snow shows high agreement between the two datasets, which is the same case for the correlations between snow and stratospheric GPH. We therefore conclude, that the information stored in the November snow cover in 20CRv2c is slightly different to the information stored in the ERA20C snow depth. **Wegmann et al. (2017)** found that Eurasian November snow depth shows much larger disagreement between 20CRv2c and ERA20C than the same snow depth in October. Moreover, rather strong centennial trends (although linear trend subtraction was done for this study) in ERA20C snow depth might impact




the running correlations. Finally, since snow depths are relatively low in October, differences
between using snow cover and snow depth might be less important from an energy transfer
point of view.
The disagreement between ERA20C and 20CRv2c may also be related to uncertainties and
inhomogeneities in both reanalyses. Many studies showed that both ERA20C and 20CRv2c are
not suitable for studies looking at trends (e.g. **Brönnimann et al, 2012; Krüger et al., 2013**)
and may include radical shifts in atmospheric circulation, particularly over the Arctic (e.g.
**Dell'Aquila et al., 2016; Rohrer et al., 2019**). **Rohrer et al. (2019)** showed that although
trends in centennial reanalyses may be spurious, at least in the Northern Hemisphere year-to-
year variability of mid-tropospheric circulation is in agreement even in the early 20th century.
Finally, although we focus here on the connection to the NAO, we did not find strong significant
correlations between autumn snow and winter WACC. As pointed out by **Peings (2019)**, the
most important driver for the WACC signal is the Ural blocking, for which we find strong
correlations throughout the 20th century (not shown).

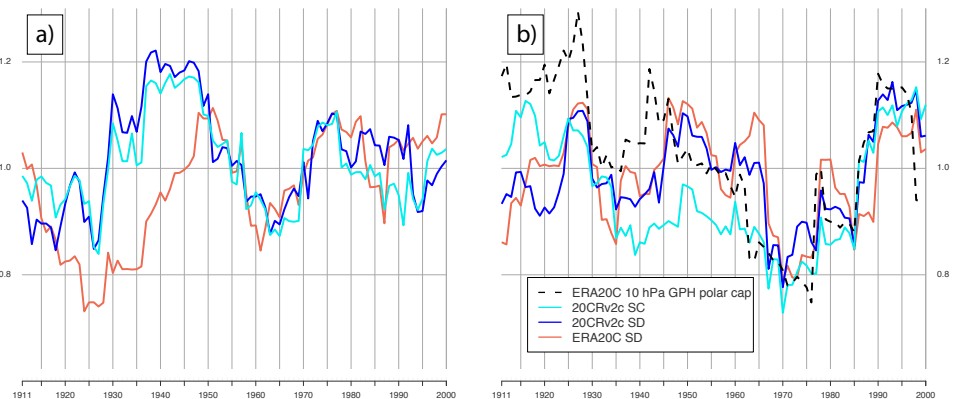


*Figure 8: 21-year running standard deviation time series of a) October snow index and b) November snow index in ERA20C and 20CRv2c (snow cover and snow depth). Dashed black line shows running standard deviation of 10 hPa November December mean GPH over the polar regions.*


## 5. Conclusion

Several reconstruction and reanalysis datasets were used to examine the link between autumn
snow cover, ocean surface conditions and the NAO pattern in winter. We found evidence for a
manifestation of a negative NAO after November with a strong west-to-east snow cover

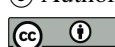



gradient, with this relationship being significant for the last 150 years. Nevertheless,
interdecadal variability for this relationship shows episodes of decreased correlation power,
which co-occur with episodes of low variability in the November snow index. This underlines
the importance of the signal-to-noise ratio for seasonal prediction studies.
Furthermore, our analysis of centennial time series supports studies pointing out the impact of
autumn snow on stratospheric circulation as well as the connection between reduced BKS ice
concentration and increased snow cover in eastern Eurasia. The latter mechanism is triggered
via the development of an atmospheric high-pressure anomaly adjacent to the BKS sea ice
anomaly, which transports moisture and cold air along its eastern flank into the continent. The
interdecadal evolution of the November snow index also points towards a co-dependence with
high North Atlantic SSTs subsequently reduced sea ice.
Extending the investigation period from 35 to 110 and up to 150 years increases the confidence
in recently proposed physical mechanisms behind cryospheric drivers of atmospheric
variability and decreases the probability of random co-variability between the Arctic
cryosphere changes and mid-latitude climate. Nevertheless, further model studies are needed to
investigate snow forcing for seasonal prediction to support the statistical links shown in this
study with causation. Future experiments should take into account year-to-year variability and
realistic distribution of snow cover if links to the stratosphere are to be examined.
For future studies regarding seasonal prediction, we emphasize the use of the November snow
dipole concerning a forecasting of the winter NAO state. Nevertheless, periods of weak
correlation might occur again, especially since it is uncertain how the sea ice to snow
relationship will change in the future, once the Arctic is ice free in summer or the local warming
is strong enough to override the counterintuitive snow cover increase. Thus, further studies are
needed to investigate the interplay between Arctic sea ice and continental snow distribution.
**Acknowledgements**
Marco Rohrer was supported by the Swiss National Science Foundation under Grant 143219.
The Twentieth Century Reanalysis Project datasets are supported by the U.S. Department of
Energy, Office of Science Innovative and Novel Computational Impact on Theory and
Experiment (DOE INCITE) program, and Office of Biological and Environmental Research
(BER), and by the National Oceanic and Atmospheric Administration Climate Program Office.
The ECMWF 20th Century Reanalyses and model simulations are supported by the EU FP7
project ERA-CLIM2.





**Data Availability**

The MERRA2 reanalysis data is publicly available at the NASA EARTHDATA repository (https://disc.gsfc.nasa.gov/daac-bin/FTPSubset2.pl). The ERA-20C reanalysis data is publicly available at the ECMWF data repository (https://apps.ecmwf.int/datasets/). The 20CRv2c reanalysis data is publicly available at the NOAA Earth System Research Laboratory repository (https://www.esrl.noaa.gov/psd/data/gridded/data.20thC_ReanV2c.html). The blocking algorithm is publicly available at https://github.com/marco-rohrer/TM2D. The AMO reconstruction data is a publicly vailable at the NOAA Earth System Research Laboratory (https://www.esrl.noaa.gov/psd/data/timeseries/AMO/). The Niño 3.4 reconstruction is publicly available at the GCOS Working Group on Surface Pressure repository (https://www.esrl.noaa.gov/psd/gcos_wgsp/Timeseries/Nino34/). The NAO reconstruction is publicly available at the Climate Research Unit repository (https://crudata.uea.ac.uk/cru/data/nao/). The Walsh et al. sea ice concentration reconstruction is publicly available at the National Snow and Ice Data Center repository (https://nsidc.org/data/g10010).

**Author Contribution**

M.W. devised the study, the main conceptual ideas and the proof outline. M.R. assisted with data availability and performed the blocking algorithm. M.W. wrote the manuscript in consultation with M.S-O. and G.L., who aided in interpreting the results.

**Competing interest**

The authors declare that they have no conflict of interest.

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

Arctic polar vortex towards the Eurasian continent in recent decades. *Nature Climate*
*Change*, *6*(12), 1094.
