# Peer review of "Eurasian autumn snow impact on winter North Atlantic"

_Earth System Dynamics, 2019_

## Referee Comment (RC1) · Anonymous Referee #1 · 23 Dec 2019

The manuscript addresses an interesting and challenging topic. Information on the relationships between Eurasian autumn snow cover and following winter North Atlantic Oscillation would be very useful for seasonal prediction. The manuscript has its merits: (a) it convincingly presents a statistical relationship between the November snow cover and winter NAO and the lack of relationship between October snow cover and winter NAO, (b) it addresses the stability of the relationships over a period of 150 years, and (c) it also pays attention to other relevant factors such as Barents – Kara sea ice cover, the Atlantic Multidecadal Oscillation, and El Nino. Further, the Introduction is very well written, demonstrating thorough knowledge on the study topic and its remaining challenges. However, the manuscript also has weaknesses, which I summarize

below. Whether the revisions needed are minor or major, depends above all on how convincingly the novelty of the results can be demonstrated (my first comment below).

Major comments

1. It should be made clearer which of the results found are novel. In the Discussion section, it is mentioned in several places that the findings support the results shown by previous studies (Gastineau et al., 2017; Han and Sun, 2018; Douville et al., 2017; Cohen et al., 2014; Wegmann et al., 2015; Yeo et al., 2016), but the novelty of the results presented remains unclear for a reader.

2. The manuscript includes parts that are carelessly written, and generate a lot of confusion.

a) In Figure 5, the projection between BKS ice concentration and November SLP anomalies shows positive values over a large region just east of Urals, but in general from the manuscript (and previous studies) I have got an impression that the decline of sea ice in BKS should favour Ural Blocking. Shouldn't this be reflected as negative projection in Figure 5b (similarly to Figure 5a)?

b) I guess that on line 350 you should refer to Figure 7e instead of Figure 7c, and make it very clear that in Figure 7e the sea ice concentration is multiplied by -1 (I guess). Also, the positive correlation seems to last until late 1960s instead of late 1970s.

c) It is not clear for me how Figure 8 supports the text on reduced variance of the snow index time series on lines 442-446. The standard deviation seems lowest in early 1900s and in 1960s.

3. The Discussion includes vague parts, such as what could be done ("doubble could" on lines 389-392), references to preliminary results not shown on lines 453-455, and lines 476-479 (this paragraph should be removed). Also, how do centennial trends impact the results, if these trends were subtracted (lines 464-466)?

Minor comments:

Line 82: remove comma

Line 112: months

Line 243: which snow indices?

Line 244: separate "and" words

Line 271: The impact does not look weak.

Line 329: Remove "slightly"

Line 421: Supplementary Figure 6

Line 428: Remove "that"
* * *

---

## Referee Comment (RC2) · Anonymous Referee #2 · 2 Jan 2020

REVIEW

Manuscript: ESD-2019-68

Title: Eurasian autumn snow impact on winter North Atlantic Oscillation depends on cryospheric variability.

Authors: Martin Wegmann, Marco Rohrer, María Santolaria-Otín and Gerrit Lohmann.

General Comments

This study presents and discusses statistical relations (diagnostics based on correlations and linear regressions) between Eurasian snow cover in autumn and wintertime

atmospheric circulation anomalies, claiming a causal link (forcing and response rela-tionship) the strength of which varies in different historical epochs. The authors make valid references to recent and past literature on this broad topic and show original and valuable results. The Reviewer would recommend this study for publication after some minor points are addressed (minor revision). In particular : (i) the authors should ac-count for serial correlation in the timeseries when assessing statistical significance, this is an important point since it can potentially affect (quite strongly) the discussed statis-tics and the associated conclusions. (ii) the authors should make an effort to be more explicit when referring to dynamical pathways, even if they do not directly assess any of the mentioned dynamical relationships (a weakness of this study). (iii) the authors should explain (otherwise remove) their line of argument on the likely driving role of ENSO in respect to low-frequency (decadal to multi-decadal) variability.

Specific Comments

1. Line 17 Perhaps the mathematical term "non-stationarity" does not convey the right message here. Obviously, predictability due to ESC varies from year to year for two basic reasons: (i) ESC anomaly may be small, thus not providing a strong forcing leading to a predictable signal, (ii) other processes affecting predictability may be more dominant.

2. Line 20 "tendency" also means time derivative. For this reason, avoid this expres-sion, or clarify.

3. Line 23 Delete "slowed"

4. Line 24 "correlation power" is not approved terminology.

5. Line 29 Three times using "power" in the abstract alone.

6. Line 34 "climate mode... over" → climate variability pattern affecting winter climate over

7. Lines 36–37 Here and elsewhere, please put a comma between "et al." and the

publication year and use semicolons to separate different references.

8. Line 38 The NAO is not defined as the strength of the gradient, it rather refers to the variability of this gradient (seesaw). Please rephrase.

9. Line 40 "its configuration" → its variability

10. Line 42 high-priority (with hyphen)

11. Line 59 manifests itself / occurs / is manifested

12. Line 79 a mechanism described by...

13. Line 89 What exactly is meant here? "forming..." how?

14. Line 93 summarized → discussed

15. Line 110 consequences → conclusions

16. Line 111 who point to the prediction power of

17. Line 114 link → chain (?)

18. Line 129 For a detailed description

19. Line 143 "found" → defined (?)

20. Line 148 The NAO centers of action are known to migrate zonally, but not so much meridionally [e.g. Barnston and Livezey (1987)].

21. Line 159 "normalized" → standardized

22. Line 165 "is above" → is higher than

23. Line 182 "the second dimension" → two dimensions (meridional and zonal direction)

24. Line 188 Blocks do not always divert the westerlies (they can also block).

25. Line 190 fulfill the two above-mentioned conditions

26. Lines 213–214 "window" → period (?)

27. Line 217 "any" → each

28. Line 233 This hints toward

29. Line 244 Please check typos (missing spaces)

30. Line 269 increase polar ("heights" is plural).

31. Line 287 "increase" → aid

32. Line 306 anomalies are regressed

33. Line 309 Remove "a" (two occurrences)

34. Line 310 "is able to support" : please rephrase

35. Line 325 "it" : please be more explicit for lucidity, what does "it" refer to?

36. Lines 327–328 "which in turn favors..." : how and why?

37. Line 329 "slightly" : this undervalues the significant differences (4 half periods vs 3 half periods, not just "slightly out of phase". In this paragraph the authors jump from an NAO reasoning to a direct connection of continental anomalies to the BKS, yet the respective dynamics are not compatible: the NAO links to more/less zonal advection, while Ural blocking links to meridional advection.

38. Lines 338–341 This approach requires a proper evaluation of the effective number of degrees of freedom, which most likely are seriously reduced due to serial correlation (related to the low-frequency nature of the discussed variability but also to the applied filter).

39. Lines 342–353 So the previously-discussed dynamics work in one decade but fail to work in another?

40. Lines 371–373 Please help the reader see whether there is anything new here in respect to the cited studies.

41. Lines 375 "popular" (is this the right word?)

42. Lines 397 low-frequency (with hyphen)

43. Line 412 ....pattern via a stratospheric pathway.

44. Line 428 Remove "that" before "seem". Referring to this paragraph, the reviewer finds the reasoning related to ENSO to be poorly based given that ENSO itself cannot be claimed to be a primary driver of (multi)decadal variability. This is an important point that should be addressed in a revised version of the manuscript.

45. Line 435 strength (not in plural)

46. Lines 433–443 Even two noisy processes after 21-year smoothing will exhibit periods of correlation and anticorrelation (purely an artifact related to limited samples and sub-samples). For robust statistics, the time window / period considered should contain at least a few periods... otherwise any result can be expected.

47. Lines 513 "counterintuitive" → contrasting (?)

FIGURE 2: How is statistical significance assessed? A suitable and rigorous test is required accounting for serial correlation (which tends to decrease the effective number of degrees of freedom). The colorbar (in this and other plots) is not a good choice as it does not allow distinguishing high from moderate values (e.g. 50 and 100 have very similar tones). Please choose a colormap with more colors. Also, add more ticks and labels in the colorbar, including the max and min values covered.

FIGURE 4: The figure caption was found in a different page (unacceptable).

FIGURE 5: The pressure unit is "Pa", not PA. Also, please define what is meant by "time unit".

[Figure]

REFERENCES: Why some appear gray and other in black font?

Please also note the supplement to this comment:
https://www.earth-syst-dynam-discuss.net/esd-2019-68/esd-2019-68-RC2-supplement.pdf

---

## Referee Comment (RC3) · Anonymous Referee #3 · 9 Jan 2020

Eurasian autumn snow impact on winter North Atlantic Oscillation depends on cryospheric variability

This study investigates the changes in the relationship between the November snow-dipole and the following winter NAO using century-long reanalyses and modern reanalysis data. The relationship between snow variability and the NAO is an important topic. The study demonstrates the correlations between the November snow-dipole, BKS sea-ice, stratospheric variability and the NAO. Using long-term reanalyses to study these correlations is a good point, although they were produced with the assimilation of limited observations. I think this is important given that most of the existing studies

are based on short temporal-range data. However, I have a few questions with the current version of the manuscript, which may be addressed by the authors.

Major comments:

1) Conclusions in this study are drawn mostly from correlations/regressions, which would affect the robustness of them. Causality is also thus hard to determine. The November Snow-dipole does have some correlations with the following wintertime NAO variability (Fig. 2). This is also true for the November BKS sea-ice (Fig. 3a). However, the physical mechanisms remain unclear since studies often contradict each other and modeling results often don't support observational relationships. I think more analyses may be considered in order to generate more convincing evidence. In addition, as argued by Peings (2019), both anomalies in the snow/sea-ice and the winter stratospheric warmings can be driven by a common driver – Ural blocking. This raises the possibility that the correlations between snow/sea-ice and the wintertime NAO are statistical ones.

2) The authors argue that the variability of the November snow-dipole largely determines the strength of the correlations between it and the wintertime NAO. But this conclusion is inferred from the 21-year running correlations and the 21-year standard deviations of the snow-dipole. The authors actually assume that the November snow-dipole is a driver of the wintertime NAO. As also mentioned in 1), causality may not be determined only from correlations/regressions.

3) The authors attribute increased correlation of the November snow-dipole (BKS sea-ice) with the wintertime NAO in recent years to the increased variability of the November snow-dipole (BKS sea-ice). Was the standard deviation of the BKS sea-ice displayed in the figures? From the analysis presented, it is hard to see how the three are correlated in a physical sense and which component of the cryosphere is more important in contributing to the recent NAO variability. There are a few studies exploring the impacts of the Arctic sea-ice on Eurasian snow. For example, Xu et al. (2019) studied

the correlation between Autumn Arctic sea-ice and the winter snow cover in Northern Eurasia.

4) I think the focus of this study needs to be clarified. The stratospheric pathway for either sea-ice or snow to impact the wintertime NAO variability is not new which can be found in many studies already cited in the introduction. Does the study emphasize the predictive nature of the correlation between the November snow-dipole and the wintertime NAO? If this is the case, why not consider some techniques such as cross-validation procedure to assess the predictive skills of the November snow-dipole? Empirical models such as those used in Chen et al. (2019; Section 6) may also be considered.

Minor comments:

1) In addition to Han and Sun (2018) and Gastineau et al. (2017), the November snow-dipole was identified in an EOF analysis by Ye and Wu (2017). 2) L28-29: Does the increased sea-ice variability enhanced that of the snow? 3) The section of Data and Methods may need some modification. In particular, more details of the reanalysis data may be given. In particular, recent satellite observations of the snow cover can be included in the analysis. 4) L153-154: In the analysis, were all the atmospheric fields detrended as well? 5) L244: Change 'aandd', 'bande' and 'candf' to 'a and d', 'b and e' and 'c and f'. 6) Labeling those multi-panel figures such as Figure 2 with additional text to indicate which variable is correlated with or regressed on to which variable may be considered to help the readers.

References: Xu, B., Chen, H., Gao, C., Zhou, B., Sun, S., & Zhu, S. (2019). Regional response of winter snow cover over the Northern Eurasia to late autumn Arctic sea ice and associated mechanism. Atmospheric Research, 222, 100-113. Chen, S., Wu, R., Song, L., & Chen, W. (2019). Interannual variability of surface air temperature over mid-high latitudes of Eurasia during boreal autumn. Climate Dynamics, 1-17. Ye, K. & Wu, R. Adv. Atmos. Sci. (2017) 34: 847. https://doi.org/10.1007/s00376-017-6287-z

---

## Author Comment (AC1) · 29 Feb 2020

The manuscript addresses an interesting and challenging topic. Information on the relationships between Eurasian autumn snow cover and following winter North Atlantic Oscillation would be very useful for seasonal prediction. The manuscript has its merits: (a) it convincingly presents a statistical relationship between the November snow cover and winter NAO and the lack of relationship between October snow cover and winter NAO, (b) it addresses the stability of the relationships over a period of 150 years, and (c) it also pays attention to other relevant factors such as Barents – Kara sea ice cover, the Atlantic Multidecadal Oscillation, and El Nino. Further, the Introduction is

very well written, demonstrating thorough knowledge on the study topic and its remaining challenges. However, the manuscript also has weaknesses, which I summarize below. Whether the revisions needed are minor or major, depends above all on how convincingly the novelty of the results can be demonstrated (my first comment below).

Major comments 1. It should be made clearer which of the results found are novel. In the Discussion section, it is mentioned in several places that the findings support the results shown by previous studies (Gastineau et al., 2017; Han and Sun, 2018; Douville et al., 2017; Cohen et al., 2014; Wegmann et al., 2015; Yeo et al., 2016), but the novelty of the results presented remains unclear for a reader.

REPLY: Thank you for your comment. We agree that the focus of this study needed to be clarified. We therefore edited the introduction and discussion part substantially to allow the reader to focus on the key messages we want to deliver.

2. The manuscript includes parts that are carelessly written, and generate a lot of confusion. a) In Figure 5, the projection between BKS ice concentration and November SLP anomalies shows positive values over a large region just east of Urals, but in general from the manuscript (and previous studies) I have got an impression that the decline of sea ice in BKS should favour Ural Blocking. Shouldn't this be reflected as negative projection in Figure 5b (similarly to Figure 5a)?

REPLY: Thank you for your comment. We realize that we forgot to mention that for Figure 5 sea ice concentration is multiplied by -1, thus Figure 5b and 5c show strong blocking together with a decline of BKS sea ice. We added that information in the figure caption of Figure 5.

b) I guess that on line 350 you should refer to Figure 7e instead of Figure 7c, and make it very clear that in Figure 7e the sea ice concentration is multiplied by -1 (I guess). Also, the positive correlation seems to last until late 1960s instead of late 1970s.

REPLY: Thanks for pointing out that mistake. We fixed the error with the Figure descrip-

tion and edited the whole paragraph accordingly. The improved description of Figure 7 can now be found from Line 361-374

c) It is not clear for me how Figure 8 supports the text on reduced variance of the snow index time series on lines 442-446. The standard deviation seems lowest in early 1900s and in 1960s.

REPLY: We reshuffled and rewrote large parts of the discussion to make the link between Arctic warm periods, increased cryospheric variability and the link to the prediction skill more apparent.

3. The Discussion includes vague parts, such as what could be done ("doubble could" on lines 389-392), references to preliminary results not shown on lines 453-455, and lines 476-479 (this paragraph should be removed). Also, how do centennial trends impact the results, if these trends were subtracted (lines 464-466)?

REPLY: Thanks for the comment. We agree that the discussion part was both incoherent and repetitive. We edited large part of the old discussion section and hopefully improved the train of thought throughout the section. We reworded the notion about the centennial trends, which are in fact not significant for the snow cover indices we use in this study (nevertheless we detrended the data just to be in line with comparable studies). What we wanted to mention are decadal trends found by Wegmann et al. 2017 for snow in long-term reanalyses. We changed the wording accordingly in lines 552-558.

Minor comments: Line 82: remove comma

REPLY: removed

 Line 112: months

REPLY: corrected

 Line 243: which snow indices?

REPLY: clarified

 Line 244: separate "and" words

REPLY: corrected

 Line 271: The impact does not look weak.

REPLY: Clarified this point

Line 329: Remove "slightly"

REPLY: removed

 Line 421: Supplementary Figure 6

REPLY: not sure what is the issue with this statement. We keep it like this for the time being.

 Line 428: Remove "that"

REPLY: removed

---

## Author Comment (AC2) · 29 Feb 2020

Eurasian autumn snow impact on winter North Atlantic Oscillation depends on cryospheric variability This study investigates the changes in the relationship between the November snow- dipole and the following winter NAO using century-long reanal- yses and modern reanal- ysis data. The relationship between snow variability and the NAO is an important topic. The study demonstrates the correlations between the November snow-dipole, BKS sea-ice, stratospheric variability and the NAO. Using long- term reanalyses to study these correlations is a good point, although they were pro- duced with the assimilation of limited observations. I think this is important given that

most of the existing studies are based on short temporal-range data. However, I have a few questions with the current version of the manuscript, which may be addressed by the authors. Major comments:

1) Conclusions in this study are drawn mostly from correlations/regressions, which would affect the robustness of them. Causality is also thus hard to determine. The November Snow-dipole does have some correlations with the following wintertime NAO variability (Fig. 2). This is also true for the November BKS sea-ice (Fig. 3a). However, the physical mechanisms remain unclear since studies often contradict each other and modeling results often don't support observational relationships. I think more analyses may be considered in order to generate more convincing evidence. In addition, as argued by Peings (2019), both anomalies in the snow/sea-ice and the winter stratospheric warmings can be driven by a common driver – Ural blocking. This raises the possibility that the correlations between snow/sea-ice and the wintertime NAO are statistical ones.

REPLY: Thank you very much for your comment. The focus on this study is not to determine causality between sea ice and snow cover. In fact other studies showed that link much better than we could here. Our study focuses on the fact that a) snow is a better predictor than sea ice and b) on the skill of the snow dipole for more than 150 years which is a novelty in the current scientific literature. We are well aware of the ongoing debate in the scientific literature about the dispute between observational studies and modeling studies. Here we argue that extending the investigation period from commonly 30 years to 150 years is important for the scientific discussion. Identifying strong relationships for 150 years is clearly a stronger argument for the existence of a physical mechanism than investigating 30 years. Our study helps to put modeling studies as well as the ongoing cryosphere changes in context. Concerning the very idealized study of Peings (2019) we mention in the discussion part the differences with our study and Peings (2019) as well as investigations we performed with blockings calculated from reanalyses. Nevertheless our study, even showing a linkage that is in line

with the physical theory of snow to stratosphere to surface climate for 150 years, can not exclude the possibility of non-causality, that is correct. We made sure to underline that aspect in the discussion part.

2) The authors argue that the variability of the November snow-dipole largely determines the strength of the correlations between it and the wintertime NAO. But this conclusion is inferred from the 21-year running correlations and the 21-year standard deviations of the snow-dipole. The authors actually assume that the November snow-dipole is a driver of the wintertime NAO. As also mentioned in 1), causality may not be determined only from correlations/regressions.

REPLY: Thank you for your comments. It is unclear to the authors were exactly the issue is with the idea that increased variability in the predictor can strengthen the statistical relationship to the predictand. We still assume that the November snow-dipole is the physical driver behind the link to the wintertime NAO, we just highlight that the change in strength of this relationship is determined by the year-to-year variability of snow cover. We agree that our wording in the discussion of Figure 8 implied causality and we changed the wording accordingly. We also restructured the Discussion section to make highlight the implications of Figure 8 (now Figure 9)

3) The authors attribute increased correlation of the November snow-dipole (BKS sea-ice) with the wintertime NAO in recent years to the increased variability of the November snow-dipole (BKS sea-ice). Was the standard deviation of the BKS sea-ice displayed in the figures? From the analysis presented, it is hard to see how the three are correlated in a physical sense and which component of the cryosphere is more important in contributing to the recent NAO variability. There are a few studies exploring the impacts of the Arctic sea-ice on Eurasian snow. For example, Xu et al. (2019) studied the correlation between Autumn Arctic sea-ice and the winter snow cover in Northern Eurasia.

REPLY: Thank you for your comments. You raised an important point. Indeed the

overlay of correlation and standard deviation was not visible. We now incorporated a new figure (Figure11) in the supplement that shows both the running standard deviation of BKS sea ice and the running correlation of BKS sea ice with the wintertime NAO. We mention it now in the discussion part. We used partial correlation to highlight the fact that snow cover is a stronger predictor for the winter NAO over long time periods that than the BKS, especially since Figure 7 shows that the BKS has a very weak relationship with the NAO for most of the 20th century. We mention the Xu et al. (2019) study and highlight that the authors looked at DJF only where as we focus on the autumn period.

4) I think the focus of this study needs to be clarified. The stratospheric pathway for either sea-ice or snow to impact the wintertime NAO variability is not new which can be found in many studies already cited in the introduction. Does the study emphasize the predictive nature of the correlation between the November snow-dipole and the wintertime NAO? If this is the case, why not consider some techniques such as cross-validation procedure to assess the predictive skills of the November snow- dipole? Empirical models such as those used in Chen et al. (2019; Section 6) may also be considered.

REPLY: Thank you very much for your comments. We agree that the focus of this study needed to be clarified. We therefore edited the introduction and discussion part substantially to allow the reader to focus on the key messages we want to deliver. As you rightly pointed out neither the stratospheric connection nor the impact on the wintertime NAO are new findings. Showing however, that these linkages are substantial and detectable for more than 100 years is a new scientific finding and an important puzzle piece for the ongoing debate that you mentioned above. Moreover highlighting the strengths of this relationship for Arctic warm periods is a new puts the current warm period in context and helps the scientific community to assess current cryopshere–atmosphere links in the framework of past climatic variations. We also newly added a very basic comparison of multiple regression prediction models based on cryosphere
predictors for the 20th century and beyond at the end of the results section (lines 396-439), which we then also discuss in the discussion section.

Minor comments: 1) In addition to Han and Sun (2018) and Gastineau et al. (2017), the November snow- dipole was identified in an EOF analysis by Ye and Wu (2017).

REPLY: Thank you for pointing out this study. We added Ye and Wu (2017) to the references.

2) L28-29: Does the increased sea-ice variability enhanced that of the snow? REPLY: There is a correlation of variability on decadal timescales especially with October snow cover, yes. It is however more non-linear than the correlation between standard deviation of snow cover and standard deviation of stratospheric polar cap height as shown in Figure 8 (now Figure 9). We added that information to the supplement.

3) The section of Data and Methods may need some modification. In particular, more details of the reanalysis data may be given. In particular, recent satellite observations of the snow cover can be included in the analysis.

REPLY: Rather than describing the snow representation of the reanalyses in this process oriented paper we refer to the studies by Wegmann et al. (2017) and Orsolini et al. (2019). If that is not enough information for the reader, we would ask the reviewer to provide specific points of information that are missing.

REPLY: From this comment it is unclear as to what information can be gained by incorporating satellite information since reliable snow cover information during by satellites is limited to the beginning of the 1980s and this study focuses on long term relationships. Nevertheless we incorporated a comparison of the Rutgers snow cover product with the reanalyses products in recent decades in Figure 2 of the new Supplementary Information and mention them in the Data & Method section.

4) L153-154: In the analysis, were all the atmospheric fields detrended as well?

REPLY: yes, we added that detail to the description of the data

5) L244: Change 'aandd', 'bande' and 'candf' to 'a and d', 'b and e' and 'c and f'.

REPLY: Changed accordingly

6) Labeling those multi-panel figures such as Figure 2 with additional text to indicate which variable is correlated with or regressed on to which variable may be considered to help the readers.

REPLY: Unclear what is meant here since it is always the same variable (DJF sea level anomalies) regressed onto the same variable (snow index).

References: Xu, B., Chen, H., Gao, C., Zhou, B., Sun, S., & Zhu, S. (2019). Regional response of winter snow cover over the Northern Eurasia to late autumn Arctic sea ice and associated mechanism. Atmospheric Research, 222, 100-113.

Chen, S., Wu, R., Song, L., & Chen, W. (2019). Interannual variability of surface air temperature over mid-high latitudes of Eurasia during boreal autumn. Climate Dynamics, 1-17. Ye, K. & Wu, R. Adv. Atmos. Sci. (2017) 34: 847. https://doi.org/10.1007/s00376-017-6287-z

---

## Author Comment (AC3) · 29 Feb 2020

This study presents and discusses statistical relations (diagnostics based on correlations and linear regressions) between Eurasian snow cover in autumn and wintertime atmospheric circulation anomalies, claiming a causal link (forcing and response relationship) the strength of which varies in different historical epochs. The authors make valid references to recent and past literature on this broad topic and show original and valuable results. The Reviewer would recommend this study for publication after some minor points are addressed (minor revision). In particular : (i) the authors should account for serial correlation in the timeseries when assessing statistical significance, this

is an important point since it can potentially affect (quite strongly) the discussed statistics and the associated conclusions. (ii) the authors should make an effort to be more explicit when referring to dynamical pathways, even if they do not directly assess any of the mentioned dynamical relationships (a weakness of this study). (iii) the authors should explain (otherwise remove) their line of argument on the likely driving role of ENSO in respect to low-frequency (decadal to multi-decadal) variability.

REPLY: Thank you very much for your comments. We address the topics for the specific comments below. We removed most of the discussion concerning the low frequency impact of ENSO and make sure to highlight the dynamical pathway more.

Specific Comments 1. Line 17 Perhaps the mathematical term "non-stationarity" does not convey the right message here. Obviously, predictability due to ESC varies from year to year for two basic reasons: (i) ESC anomaly may be small, thus not providing a strong forcing leading to a predictable signal, (ii) other processes affecting predictability may be more dominant.

REPLY: Nonstationarity appears to be a common phrasing in climate science for the time dependance of the predictor to the predictand (see e.g. Kolstad & Screen 2019) and as such we keep this phrase for now but are open to specific suggestions.

2. Line 20 "tendency" also means time derivative. For this reason, avoid this expression, or clarify.

REPLY: We changed the wording to "NAO-like impact" throughout the document

3. Line 23 Delete "slowed"

REPLY: deleted

4. Line 24 "correlation power" is not approved terminology.

REPLY: We changed the wording to "strength"

5. Line 29 Three times using "power" in the abstract alone.

[Figure]

REPLY: We changed the wording to "value"

6. Line 34 "climate mode... over" → climate variability pattern affecting winter climate over

REPLY: We changed the wording

7. Lines 36–37 Here and elsewhere, please put a comma between "et al." and the publication year and use semicolons to separate different references.

REPLY: Changed accordingly

8. Line 38 The NAO is not defined as the strength of the gradient, it rather refers to the variability of this gradient (seesaw). Please rephrase.

REPLY: Rephrased accordingly

9. Line 40 "its configuration" → its variability

REPLY: Changed accordingly

10. Line 42 high-priority (with hyphen)

REPLY: Changed accordingly

11. Line 59 manifests itself / occurs / is manifested

REPLY: Rephrased accordingly

12. Line 79 a mechanism described by...

REPLY: Rephrased accordingly

13. Line 89 What exactly is meant here? "forming..." how?

REPLY: Specified the dynamic linkage

14. Line 93 summarized → discussed

REPLY: Rephrased accordingly

15. Line 110 consequences → conclusions

REPLY: Changed accordingly

16. Line 111 who point to the prediction power of

REPLY: Changed accordingly

17. Line 114 link → chain (?)

REPLY: rephrased accordingly

18. Line 129 For a detailed description

REPLY: Rephrased accordingly

19. Line 143 "found" → defined (?)

REPLY: Rephrased accordingly

20. Line 148 The NAO centers of action are known to migrate zonally, but not so much meridionally [e.g. Barnston and Livezey (1987)].

REPLY: Deleted the mentioning of NAO and instead replaced with "jet"

21. Line 159 "normalized" → standardized

REPLY: Rephrased accordingly

22. Line 165 "is above" → is higher than

REPLY: Rephrased accordingly

23. Line 182 "the second dimension" → two dimensions (meridional and zonal direction)

REPLY: Rephrased accordingly

24. Line 188 Blocks do not always divert the westerlies (they can also block).

REPLY: Rephrased accordingly

25. Line 190 fulfill the two above-mentioned conditions

REPLY: Rephrased accordingly

26. Lines 213–214 "window" → period (?)

REPLY: Rephrased accordingly
 27. Line 217 "any" → each

REPLY: Rephrased accordingly

28. Line 233 This hints toward

REPLY: Rephrased accordingly

29. Line 244 Please check typos (missing spaces)

REPLY: Corrected

30. Line 269 increase polar ("heights" is plural).

REPLY: Corrected

31. Line 287 "increase" → aid

REPLY: Rephrased accordingly
32. Line 306 anomalies are regressed

REPLY: Corrected

33. Line 309 Remove "a" (two occurrences)

REPLY: Corrected

34. Line 310 "is able to support" : please rephrase

REPLY: Rephrased accordingly

35. Line 325 "it" : please be more explicit for lucidity, what does "it" refer to?

REPLY: Clarified and extended the sentence

36. Lines 327–328 "which in turn favors..." : how and why?

REPLY: Added additional information for the reader

37. Line 329 "slightly" : this undervalues the significant differences (4 half periods vs 3 half periods, not just "slightly out of phase"). In this paragraph the authors jump from an NAO reasoning to a direct connection of continental anomalies to the BKS, yet the respective dynamics are not compatible: the NAO links to more/less zonal advection, while Ural blocking links to meridional advection.

REPLY: We removed the slightly notion and clarified the train of thought for the connection between the BKS and the continental anomalies (lines 340-352).

38. Lines 338–341 This approach requires a proper evaluation of the effective number of degrees of freedom, which most likely are seriously reduced due to serial correlation (related to the low-frequency nature of the discussed variability but also to the applied filter).

REPLY: We addressed the question of serial correlation by performing Durbin-Watson tests for every pair shown in Figure 7 and did not find compelling evidence for the existence of serial correlation in these relationships. We added that information to the text (lines 384-389) and show the Durbin-Watson test statistics in the Supplement.

39. Lines 342–353 So the previously-discussed dynamics work in one decade but fail to work in another?

REPLY: Thank you for your comment. We are not able to exclude the possibility that the dynamics (as in physical mechanism) are still working during times with weak correlation. However, the mechanism might be weaker due to reduced variability in the predictor. Therefore we focus here on the statistical strength of this relationship rather than excluding the possibility of the mechanism still being the same mechanism, even in times of low correlation.

40. Lines 371–373 Please help the reader see whether there is anything new here in respect to the cited studies.

REPLY: We edited the introduction and discussion part substantially to allow the reader to focus on the key messages we want to deliver and to highlight new findings.

41. Lines 375 "popular" (is this the right word?)

REPLY: deleted

42. Lines 397 low-frequency (with hyphen)

REPLY: Changed accordingly

43. Line 412 . . ..pattern via a stratospheric pathway.

REPLY: We add information about this in the discussion (lines 464-474)

44. Line 428 Remove "that" before "seem". Referring to this paragraph, the reviewer finds the reasoning related to ENSO to be poorly based given that ENSO itself cannot be claimed to be a primary driver of (multi)decadal variability. This is an important point that should be addressed in a revised version of the manuscript.

REPLY: We agree with the reviewer that the ENSO discussion is weak and not helping the focus of this manuscript. We therefore deleted this paragraph.

45. Line 435 strength (not in plural)

REPLY: Deleted this sentenced and moved the necessary information to the beginning of the discussions section

46. Lines 433–443 Even two noisy processes after 21-year smoothing will exhibit periods of correlation and anticorrelation (purely an artifact related to limited samples and sub-samples). For robust statistics, the time window / period considered should contain at least a few periods... otherwise any result can be expected.

REPLY: Thank you for your comment. It is not entirely clear for the authors what is

meant by considering different time windows for which analysis. However we show now in the Supplement a new Figure 8, which is basically Figure 7b for different running correlation windows ranging from 5 years to 31 years. We find that the main outcome of the analysis is not dependent on the time window changing from 11 to 21 to 25 to 31 years. The 5year window is noisy due to the nature of the high frequency variability in the system. These findings are consistent between ERA20C and 20CRv2c. We also added that information to the text (lines 384-389).

47. Lines 513 "counterintuitive" → contrasting (?)

REPLY: We stay with counterintuitive but now it refers to "anthropogenic global warming"

FIGURE 2: How is statistical significance assessed? A suitable and rigorous test is required accounting for serial correlation (which tends to decrease the effective number of degrees of freedom). The colorbar (in this and other plots) is not a good choice as it does not allow distinguishing high from moderate values (e.g. 50 and 100 have very similar tones). Please choose a colormap with more colors. Also, add more ticks and labels in the colorbar, including the max and min values covered.

REPLY: Thank you for your comment. We now assess auto-correlation of the ERA20C snow index time series, the 20CRv2c snow index time series, the ERA20C 10hPa GPH time series and the BKS time series with auto-correlation function plots in Supplement Figure 5. We found no clear auto-correlation signal in the snow and stratosphere, however we found auto-correlation in the BKS sea ice index. We now highlight that fact in the text concerning Figure 2 to make the reader aware of that issue. Moreover, we found auto-correlation for the AMO index (as expected for) and no auto-correlation for the ENSO index. We do not show that information in the Supplement but we mention it in the manuscript text. Furthermore we updated the colormaps for Figures 2&3 and 5b with a higher range of values as well as higher color step change resolution.

FIGURE 4: The figure caption was found in a different page (unacceptable).

REPLY: Moved up the figure caption.

FIGURE 5: The pressure unit is "Pa", not PA. Also, please define what is meant by "time unit".

REPLY: Clarified and corrected

REFERENCES: Why some appear gray and other in black font?

REPLY: That seems to be an artifact of the conversion process to pdf. We double checked and hope to have fixed that issue.

---

## Author Response (AR2)

Reviewer #3

Eurasian autumn snow impact on winter North Atlantic Oscillation depends on cryospheric variability

The manuscript has been improved. I have two remaining issues that may be addressed by the authors.

1) If the purpose of the study is not to determine any sort of causality, then I think the manuscript title should be changed to reflect this. I also think the authors should check through the text to ensure that causality is not inferred.

**REPLY: Thank you very much for your comment. In order to reduce the amount of inferred causality linguistically, we changed the title of the manuscript to Eurasian autumn snow link to winter North Atlantic Oscillation strongest for Arctic warming periods. Moreover, we changed specific wording like „impact" or „connection" throughout the document, especially so in the discussion section.**

2) Conclusions from Fig. 6 are quite speculative. I don't know what critical information this figure has conveyed. Does this figure support the key messages?

**REPLY: Thank you very much for your comment. Indeed, the value of the information in terms of dynamics is not very high for Figure 6. However, we think it has some value in conveying the time evolution of different climate indices over the last 130 years in a very quick and convenient way, rather than just explaining that time evolution in text. Nevertheless, we deleted the time series for the ENSO evolution, since we do not discuss that relationship in detail. With this, the Figure is more concise and acts as a visual guide for the reader. As such, we decided to keep Figure 6 in the manuscript.**

Reviewer #2

The authors have well taken into accoint my comments. I sugest acceptance of the manuscript after the following technical corrections:

Line 86: more powerful than what? I guess sea ice, but make it clear.

**REPLY: Added information accordingly. Yes, we meant sea ice.**

Line 100: seem

**REPLY: Fixed**

Line 119: snow cover

**REPLY: Fixed**

Line 449: onwards: use past tense systematically when describing what was done.

**REPLY: We now use past tense systematically for the discussion section**

Line 502: impacted by what?

**REPLY: Added explanation as to what is impacted by which phenomenon**

Line 557: drop comma

**REPLY: Fixed**

Line 600: drop "a"

**REPLY: Fixed**

List of relevant changes:

1) Title change
2) Exchanged vocabulary that supported causality

[revised manuscript text omitted]